

# Global meta-analysis of water cycling response to plant mixture

Huaqing Liu[1,2,5], Xiaodong Gao[1,2,5], Changjian Li[2,3], Yaohui Cai[2,3], Xiaolin Song[4], Xining Zhao[1,2,5] *

[1] The Research Center of Soil and Water Conservation and Ecological Environment, Chinese Academy of Sciences and Ministry of Education, Yangling, Shaanxi 712100, China

[2] Institute of Soil and Water Conservation, Chinese Academy of Sciences and Ministry of Water Resources, Yangling, PR China

[3] National Engineering Research Center of Water Saving and Irrigation Technology, No. 26, Xinong Road, Yangling, 712100, Shaanxi Province, China

[4] College of Horticulture, Northwest A&F University, Yangling, Shaanxi, 712100 China

[5] University of Chinese Academy of Sciences, Beijing, 100049, China

*Correspondence to*: Xining Zhao (zxn@nwafu.edu.cn)

**Abstract.** Plant mixtures maintain irreplaceable ecosystem services through biodiversity conservation, guarantee food security, and buffer climate change. However, the effects of plant mixing on the water cycle remain poorly understood despite the significance of the water cycle in maintaining life on Earth. Here, we conducted a meta-analysis using a global data set of

1,631 paired observations from 88 publications to explore how plant mixtures affect seven critical water cycle processes (soil water content, runoff, infiltration, soil evaporation, canopy transpiration, throughfall, and water use efficiency). We found that plant mixtures reduced mean soil water content (1.5%), runoff (12.3%), and soil evaporation (7.8%), while improving soil infiltration (42.7%), leaf transpiration (13.5%), and water use efficiency (40.9%). The effect size of plant mixtures greatly depended on ecosystem types; crop mixtures significantly reduced soil evaporation (10.7%) and runoff (22.1%), whereas

mixed forest and agroforestry systems significantly enhanced soil infiltration rates. Moreover, the effect size of the plant mixture also varied with climate (mean annual precipitation), soil properties (organic matter content, bulk density, and total nitrogen content), and management practices (crop type, fertilization, and irrigation). In plant minutes, resource complementarity, abiotic facilitation and biotic feedback may be the underlying mechanisms that regulate water cycle. This work highlights the importance of plant mixtures in facilitating positive water cycles and provide insights into the

establishment of sustainable artificial ecosystems.

## 1 Introduction

A sustainable water cycle is essential for ecosystems and society to persist and thrive. Plants are the main biological regulators of regional and global water cycles and determine the hydrological fluxes of ecosystems (Deng et al., 2017; Lemordantn et al., 2018). Human activities, such as deforestation and agriculture have profoundly changed the community structure, number, and

distribution of plants both regionally and globally, resulting in a great loss of biodiversity. Extensive studies have demonstrated a strong negative impact of plant diversity loss on ecosystem nutrient and carbon cycles (Duffy et al., 2017; Feng et al., 2022; Li et al., 2020; Prommer et al., 2020; Timaeus, Weedon, & Finckh, 2022) and microclimate conditions (Hofmeijer et al., 2021).



The impact of plant diversity loss on ecosystem function has profoundly changed regional and global water cycling (Levia et al., 2020), and precluded the realization of the 15th Sustainable Development Goal (SDG 15). Plant mixtures where two or more species coexist not only promote species recruitment, but have the potential to maintain the water cycle (Lange et al., 2019).

Plant mixtures regulate water cycle processes by modifying substrate characteristics and regulating the vertical movement/distribution of water. (Lange et al., 2019; Nyawade et al., 2019; Zheng et al., 2018). Plant mixtures increase the complexity of the substrate by aerating the soil, increasing biomass, and changing ground cover characteristics among other effects (Barry et al., 2019; Zhu et al., 2015). This affects the distribution of precipitation between the above- and below-ground components and has a wide range of effects on the water cycle, such as evaporation, stemflow, throughfall, soil infiltration, and surface runoff (Collins & Bras, 2007; Luo et al., 2022; Li et al., 2022; Unigarro et al., 2023). Although extensive research has demonstrated the potential of plant diversity to benefit the water cycle, potential negative effects have also been highlighted. For example, some studies have demonstrated that overproduction of mixture relative to monoculture reduce throughfall while increasing stemflow, improving infiltration and soil water availability (Göransson et al., 2016; Leimer et al., 2018). Although an increase in biomass also promotes leaf transpiration (Verheyen et al., 2008), an increase in plant cover might compensate for increased transpiration by reducing soil evaporation (Fischer et al., 2019). Additionally, most studies have focused only on the effects of plant mixtures on individual or several of these water cycle processes (TABLE 1). Soil moisture has received the greatest attention among water cycle processes as the link between the above- and below-ground hydrological cycles (Moyano et al., 2013). However, according to the region, studies have found contradictory (Rahman et al., 2017; Guderle et al., 2018) or negligible (O'Keefe et al., 2019) effects of plant mixtures on soil water content as well as for other water cycle processes ( Ghahremani et al., 2021; Leimer et al., 2014; Spehn et al., 2000). It is difficult to accurately and comprehensively assess the impact of plant mixture on the overall water cycle by evaluating their effects on individual water cycle processes (Barry et al., 2020). Therefore, it is necessary to conduct a systematic and quantitative study of the effects of plant mixtures on the water cycle on a global scale that considers multiple water cycle processes.

Although no global-scale studies have been conducted to determine the causes of the differential effects of plant mixtures on water cycle processes, several review articles propose that this variability may depend on the type of plant mixture, planting period, soil properties, climate, and management practices (Augusto & Boča, 2022; Feng et al., 2022; Freschet et al., 2017; Mori et al., 2020; Toïgo et al., 2022; Wright et al., 2017). These factors also influence vegetation growth (Guderle et al., 2018) and soil physicochemical properties (Furey & Tilman, 2021). For example, deep roots in the mixed grasslands allow soil moisture to be exhaust utilized at vertical depth (Kulmatiski et al., 2020), and may also promote drainage in agroforestry (Wu et al., 2020). Plant diversity increases soil organic matter content, which is positively correlated to water holding capacity (Lange et al., 2015). The scarcity of studies on the effects of these factors on the water cycle has hindered and in-depth understanding of the mechanisms by which plant mixtures affect water cycle, which is critical for studying the relationship between biodiversity and productivity.



**Table 1** Effect of plant mixtures on various water cycle processes.

| Water cycle process | Mixture effect | | |
| --- | --- | --- | --- |
| | Positive | Negative | Neutral |
| Soil water content | (Rahman et al., 2017), (Fan et al., 2016) | (Gong et al., 2020), (Guderle et al., 2018) | (Kimberly, Jesse, & Katherine, 2019; Wang et al., 2017) |
| Runoff rate | (Ding et al., 2015) | (Machiwal et al., 2021) | (Jiang et al., 2007) |
| Evaporation | (Chen & Zheng, 2018) | (Fan et al., 2016) | (Gao et al., 2008) |
| Infiltration rate | (Tetteh et al., 2019) | | (Ghahremani et al., 2021) |
| Transpiration | (Chen et al., 2016) | (Wang et al., 2017) | (Xiong et al., 2016) |
| Throughfall | (Luo et al., 2004) | (Rahman et al., 2017) | |
| Water use efficiency | (Nyawade et al., 2019) | (Gomes et al., 2014) | |

To elucidate the overall effect of plant mixtures on the water cycle, we conducted a meta-analysis using 1,631 paired observations from 88 publications. Our main objectives were to address: (i) determine whether plant mixtures have a positive impact on the water cycle, regardless of the region, and (ii) elucidate the underlying mechanism and roles of several influencing factors. We first examined the overall effects of plant mixtures on soil water content, runoff, infiltration, soil evaporation, leaf transpiration, throughfall, and water use efficiency. Second, we tested the impact of plant stand age, climate, initial soil physicochemical properties, and management practices on the performance of plant mixtures versus monocultures. Finally, we reviewed the existing research to determine the underlying mechanisms of the impact of plant mixtures on water cycle processes.

## 2 Materials and method

### 2.1 Data collection

We identified peer-reviewed papers published up to 1 Jun 2022 in the Web of science, SCOPUS, and China Knowledge Resource databases using the following combinations of terms: biodiversity, plant diversity, species richness, species diversity, species abundance, monoculture, mixture, or intercropping with water fluxes, water cycle, runoff, infiltration, evaporation, evapotranspiration, transpiration, soil water, soil moisture, or interflow. The following criteria were applied for the selection of publications: (a) values for water cycle process were available in the text, tables, and/or figures; (b) genotype mixtures with species were not included; (c) each plant mixture was compared to the corresponding monocultures. Data in the figures were extracted using Web Plot Digitizer (https://automeris.io/WebPlotDigitizer; Burda et al., 2017).

A total of 115 studies with 1,631 paired observations from 88 publications were selected for this meta-analysis (FIGURE 1, Supporting Information FIGURE S1, TABLE S1). Different localities within the same study were considered independent



studies. For each study, we extracted soil volumetric water content (SWC) at different soil depths. If the volumetric water content was not provide, these were calculated using the bulk density and gravity water content. Water cycle process data were

also collected, including runoff rate (RO), infiltration rate (IR), soil evaporation (E), throughfall (Th), leaf transpiration (LT), and water use efficiency (WUE). The definitions of each variables and its unit are shown in Table 2. Species richness, stand age, species proportions, plant patterns, water supply, fertilizer, and legumes were extracted from the original publications.

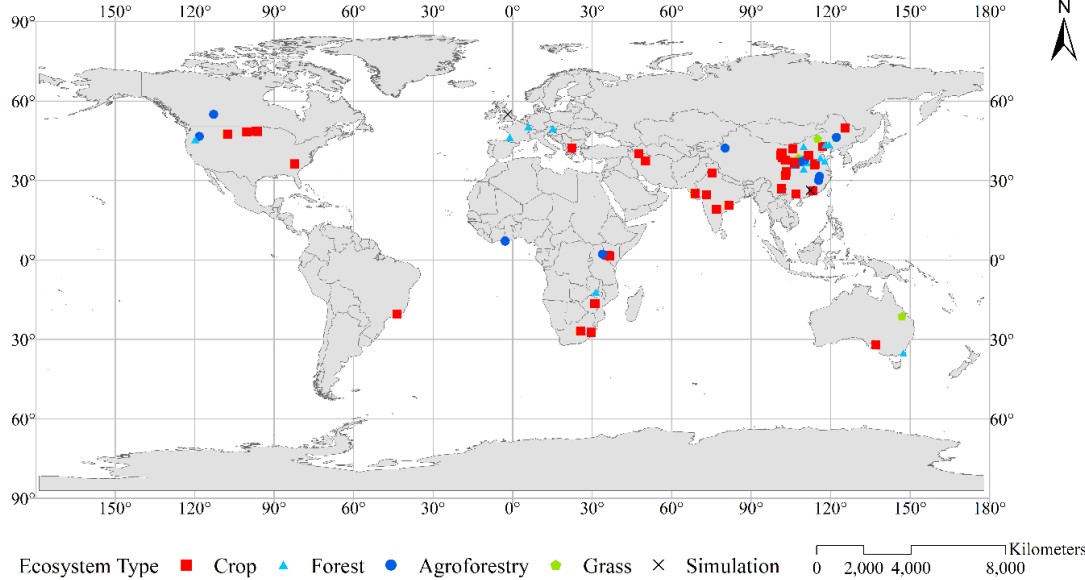

**Figure 1:** Global distribution of plant mixtures experiments focusing on diversity effects on water cycle processes in this meta-analysis.


**Table 2** Water cycle processes definitions and units.

| Water cycle process | Definitions | Units |
| --- | --- | --- |
| Soil water content (SWC) | Soil volumetric water content. | $cm^{-3}\ cm^{-3}$ |
| Runoff (RO) | Surface water flow when the ground is saturated or the rainfall intensity exceeds the infiltration rate. | mm |
| Evaporation (E) | Water evaporation from the soil surface into vapour. | mm |
| Infiltration rate (IR) | Flow rate of water that enters topsoil. | $mm\ min^{-1}$ |
| Transpiration (LT) | Water that has been taken up by plants and transpired onto the leaf surface. | $mmol\ m^{-2}\ s^{-1}$ |
| Throughfall (TF) | The amount of rainfall that is not intercepted by the crown canopy and reaches the soil. | mm |
| Water use efficiency (WUE) | The amount of biomass produced per unit volume of water evapo-transpired. | $g\ m^{-2}\ mm^{-1}$ |



Basal areas or densities in forests and agroforestry, seeds sown in grasslands, and the number of individuals in the simulation
were extracted as species proportions. In cropland, the area occupied by different species was used instead, as the effects of
different seeding methods were considered. The average value of the measurement interval was converted to the soil moisture
depth. The determination of plant stand age in different ecosystems was consistent with that reported by Peng & Chen (2021).
Ecosystem types were categorized as forests, croplands, agroforest, grasslands, and indoor plantation. We obtained 20
environmental factors from the original papers, including 1) topographic (altitude, latitude, longitude), 2) climate (mean annual
precipitation, mean annual temperature), 3) soil properties (soil organic carbon [SOC, including soil organic matter], nitrogen,
available nitrogen, phosphorous, available phosphorous, potassium, available potassium, sand, silt, clay, pH, bulk density), 4)
management practices (water application type [rainfed or irrigation], seeding pattern [addition or substitution], fertilizer use
[with or without], crop type [legume or non-legume]). We extracted missing of MAT and MAP values from WorldClim v2
(http://www.worldclim.com/version2) database according to the geographic location of the study site noted in the
corresponding article (Fick & Hijmans, 2017). Missing values of elevation were searched by latitude and longitude using Free
Map Tools (https://www.freemaptools.com/elevation-finder.html), and latitude by site name or location using Find Latitude
and Longitude (https://www.findlatitudeandlongitude.com/). In the combined dataset, most sampling depths for soil water
content were ay 0–100 cm, and the maximum was 500 cm in forests. Most soil water content data were collected from to 0–2
years of crop mixtures, and the maximum was 61 years for forest mixtures.

## 2.1 Data analysis

Log response ratios (lnRR) were used as a measure of effect size (Hedges, Gurevitch, & Curtis, 1999):

$$\ln RR = \ln(\frac{X_t}{X_c}) \tag{1}$$

where Xt is the observed value in the mixture and Xc is the expected value, which is the weighted mean of the corresponding
monoculture according to the proportion of species in the mixtures (Loreau & Hector, 2001; Peng & Chen, 2021). The
proportion of species in the mixtures was defined in terms of the individual frequency or cover area (Grossiord et al., 2013).
As many previous studies, the number of replications was employed to calculate the estimate weight (Pittelkow et al., 2015).

$$Wr = \frac{(N_c \times N_t)}{(N_c + N_t)} \tag{2}$$

where Wr is the weight for observed values and $N_t$ and $N_c$ are the number of replications in mixtures and monocultures,
respectively.

The linear, log-linear and quadratic functions for soil depth (D) and stand age (A) were compared to determine which
expression conformed to the assumption of linearity for each water cycle process based on the lowest Akaike's information
criterion (AIC) (Peng & Chen, 2021). Similar to previous meta-analyses (Mori et al., 2020), we used species richness as a
random term because of limited data (more than 90% of the studies uses used two-species mixtures).



The following model was employed to test the effects of D and A on the lnRR of SWC, RO, IR, E, Th, LT, and WUE:

$\ln RR = \beta_0 + \beta_1 \times D + \beta_2 \times A + \beta_3 \times D \times A + \pi_{study} + \varepsilon$ (3)

where β_s are the coefficients to be estimated, π_study is the random effect factor of the study, and ε is the sampling error. We derived the most parsimonious model based on the lowest AIC to prevent overfitting. As for IR, LT, Th, and WUE, we selected the null models because they had the lowest AIC values.

All continuous values were scaled for analyses. LnRR and its corresponding 95% confidence interval were transformed to
percentages using the following equation:

$(e^{\ln RR} - 1) \times 100\%$ (4)

Whether the 95% confidence intervals (CIs) crossed zero was used as the criterion for significance.

Funnel tests and treatment response ratios (against sample size) were calculted to predict publication bias. There is no publication bias in any test.

All analyses were performed in R 4.1.0 (R Core Team, 2020). Model selection was conducted using the "dredge" function of the *MuMln* package (Bartoń, 2020). The restricted maximum likelihood estimation method of the *lme4* package was implemented for all analyses, except for the bias analysis which used the restricted likelihood estimation method (Bates et al., 2015).





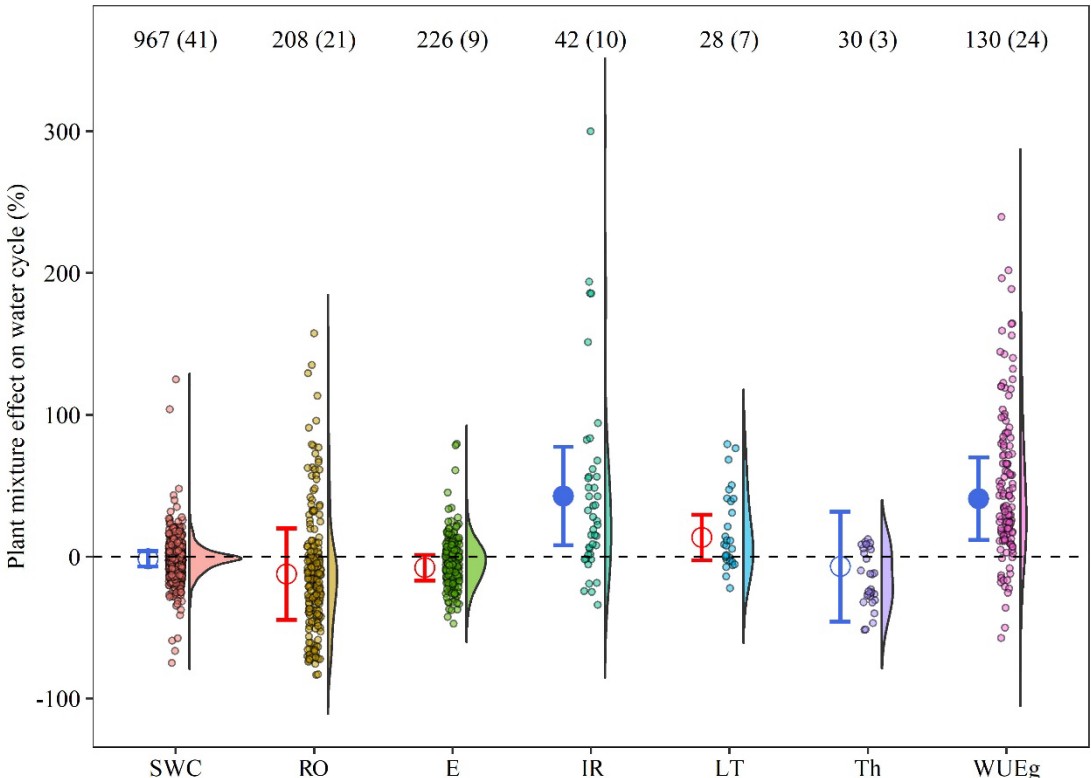

**Figure 2:** Comparison of water cycle in plant mixture and monocultures. The blue color indicates the water cycle process of the input terrestrial ecosystem, and the red color represents the water cycle process of the output ecosystem. Solid circles mean significant effect ($P <$ 0.05), hollow circles mean no significant effect. The numbers outside the parentheses indicated the number of observations pairs, while inside the parentheses indicated the number of studies.

## 3 Results

### 3.1 Overall effect of plant mixture on water cycle

Our results confirmed that plant mixtures help facilitate a positive water cycle at the local scale. On average, plant mixtures showed reduced SWC (95% CI, -6.92–3.92%), RO (95% CI, -44.59–19.97%), and E (95% CI, -16.81–1.25%) compared to monocultures, whereas IR (95% CI, 7.92–77.47%, P < 0.05), LT (95% CI, -2.45–29.43%), and WUE (95% CI, 11.79–69.96%, P < 0.05) were increased (FIGURE 2). The effect of plant mixtures on SWC decreased with soil depth (P < 0.001) but not with plant stand age (P = 0.101) (FIGURE 3). The effect of the plant mixture on E, IR, and RO increased with stand age, whereas WUE, LT, and Th decreased (FIGURE S2).



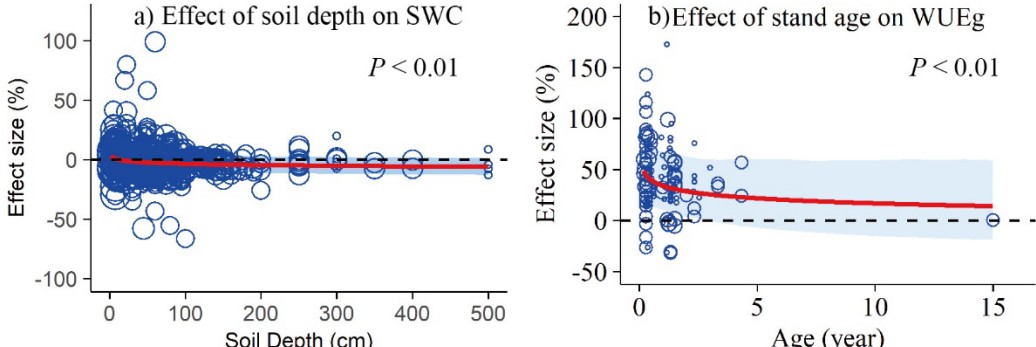

**Figure 3:** Plant mixture effect on the soil water content in terms of soil moisture measurement depth and plant stand age. (a) soil depth, (b) plant stand age. The red lines are fitted effect sizes, with bootstrapped 95% confidence intervals shaded in blue. The size of circles (Wr) represents the relative weights of corresponding observations.

## 3.2 Water cycle response of various plant mixture ecosystem

The effect size of plant mixtures on the water cycle was similar between ecosystem types, except for croplands (FIGURE 4). LT and WUE varied considerable between croplands and otherecosystem types. Furthermore, the crops were grown for a shorter period of time; except for LT, the number of observations accounted for >50% (TABLE S5). Therefore, the effect of crop diversification on water cycle was analyzed separately.



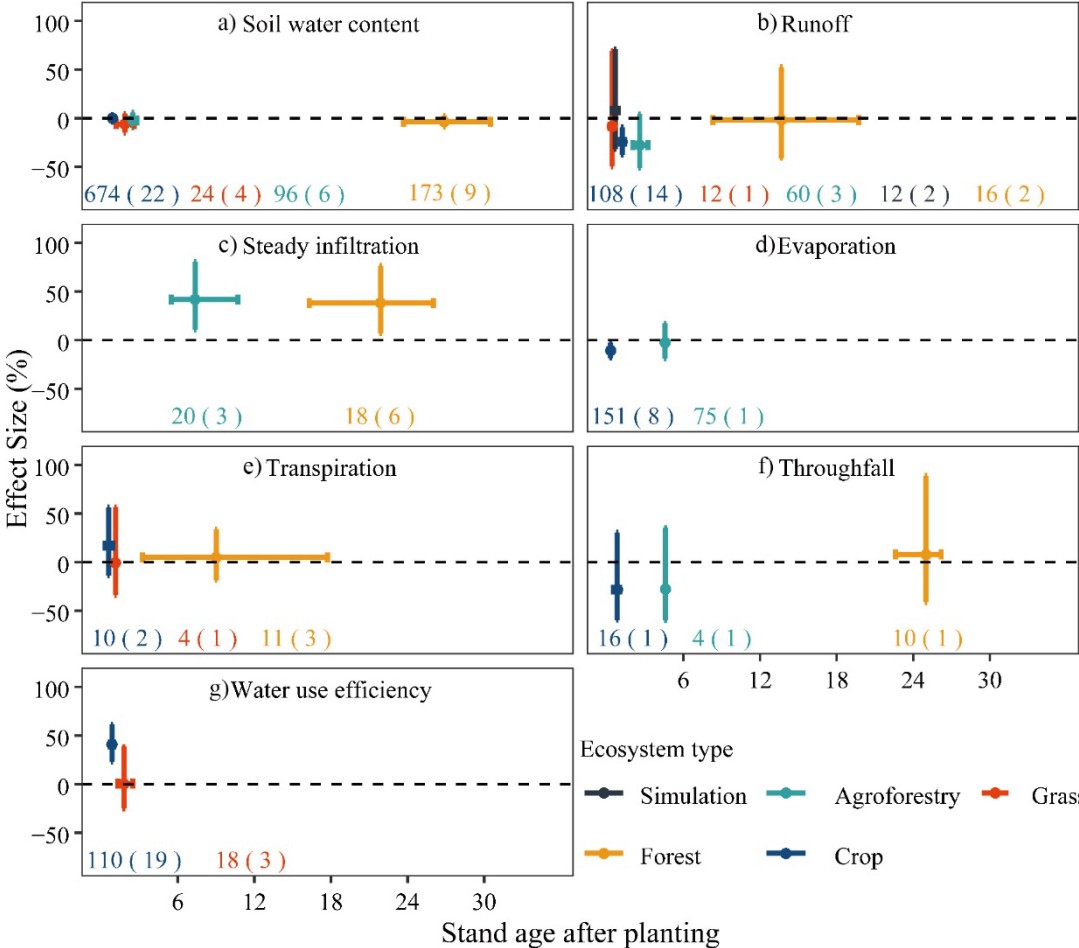

**Figure 4:** Comparison of SWC, RO, IR, E, LT, Th, and WUE in diversification assemblages versus monocultures between ecosystem types. Means and vertical and horizontal error bars represent means and 95% confidence intervals (CIs) for plant mixture effects and stand age in mixtures. The numbers outside the parentheses indicated the number of observations pairs, while inside the parentheses indicated the number of studies.

The results showed that crop mixture significantly reduced RO by 15% (95% CI, -37.51% – -6.66%, P < 0.05) and E by 7% (95% CI, -19.21% – -2.21%, P < 0.05), and increased WUE by 50% (95% CI, 18.88%–70.86%, P < 0.05) (FIGURE S3). In the cropland mixtures, the maximum sampling depth of SWC was 160 cm, and the maximum stand age was 2.3 years. The effect of crop-plant mixtures on SWC decreased with soil depth (P < 0.001) but increased with stand age (P < 0.001) (FIGURE 5). Although there were no significant effects on E, RO, LT, or WUE, the effect size decreased with crop stand age (FIGURE S4).



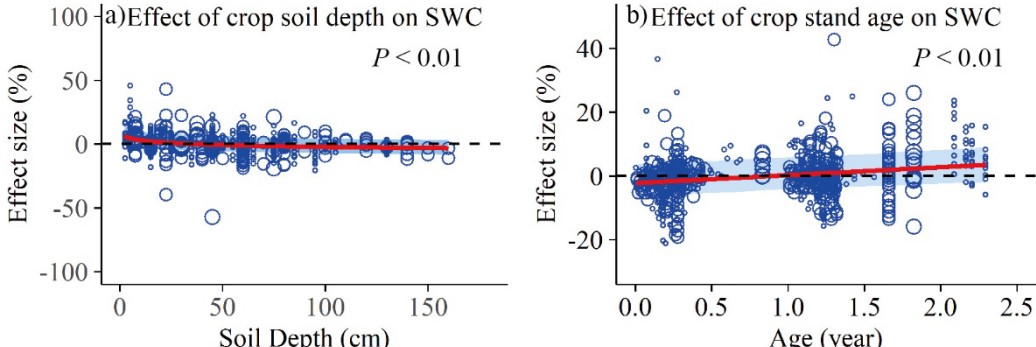

**Figure 5:** Crop mixture effect on soil water content in terms of soil moisture measurement depth and plant stand age. (a) soil depth, (b) plant stand age. The red lines are fitted effect sizes, with bootstrapped 95% confidence intervals shaded in blue. The size of circles (Wr) represents the relative weights of corresponding observations.

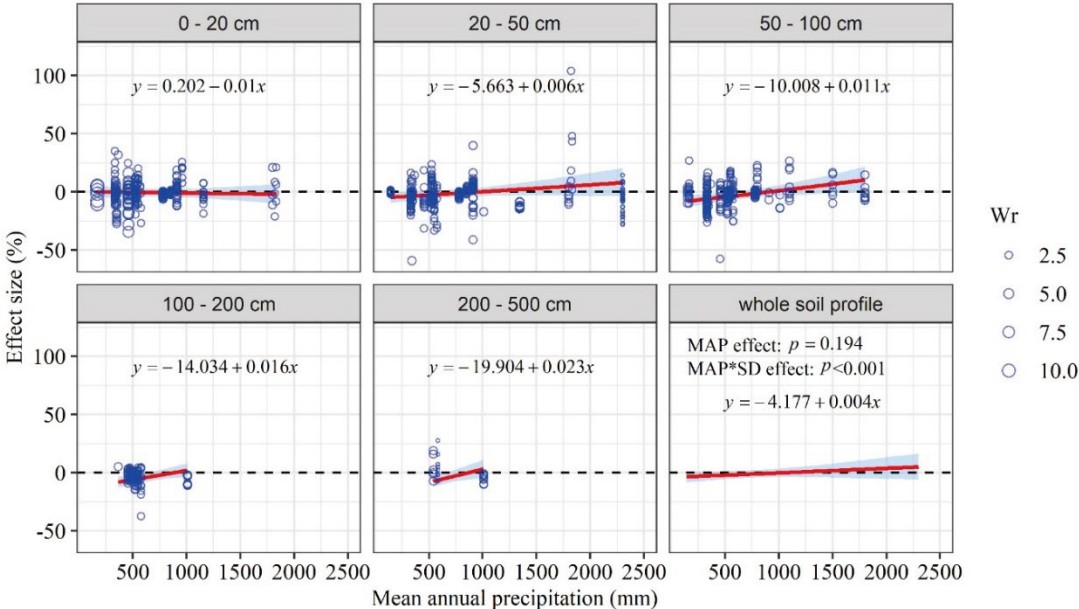

**Figure 6:** Interactive effects of mean annual precipitation (MAP) and soil moisture measurement depth (SD) on the effect size of plant mixture on soil water content. The red line represents the estimated mean response, with bootstrapped 95% confidence intervals shaded in blue. The figure was plotted based on the most parsimonious models derived from Equation 7. The size of circles (Wr) represents the relative weights of corresponding observations.

### 3.3 Effect of climate on water cycle in plant mixtures

Overall, the effect of plant mixtures on soil water content increased with MAP (P = 0.019), whereas E and Th decreased (P = 0.006 and 0.005, respectively). The effect of MAP on SWC varied significantly at different soil depths (FIGURE 6). The effect of plant mixture on SWC increased slightly with MAP in the topsoil (0-20 cm) in deeper soil. MAT had a significant effect on SWC with soil depth (MAT×SD, P < 0.001) (FIGURE 7). The effect of the plant mixture change from negative to positive





with both MAP and MAT. The results also demonstrated the negative relationship between the effect size of plant mixtures on E and Th with MAP and MAT (FIGURE 8).

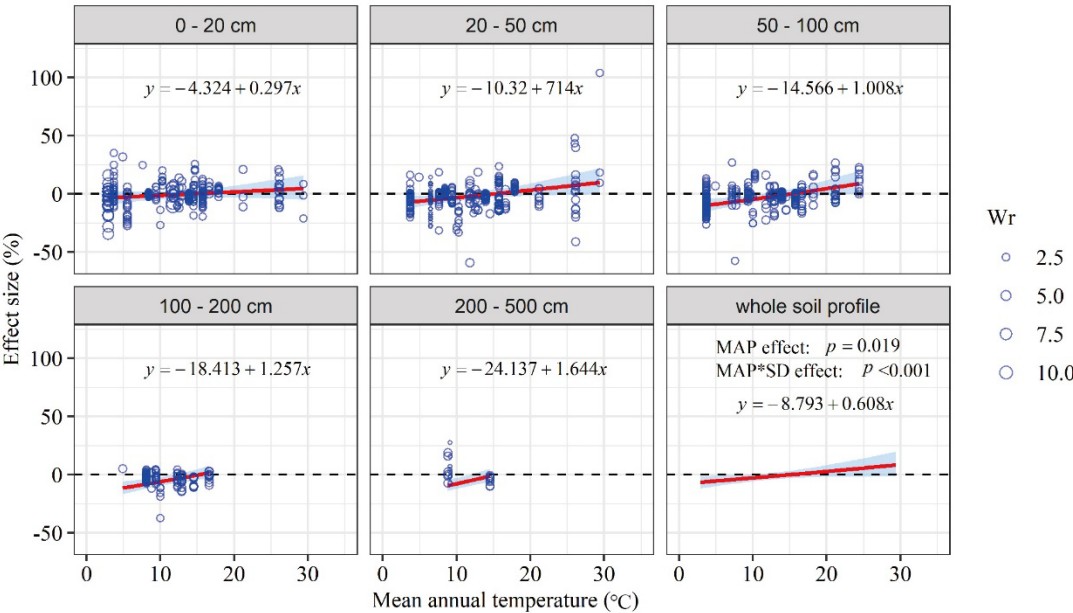

Figure 7: Interactive effects of mean annual temperature (MAT) and soil depth (SD) on the effect size of plant mixture on soil water content. Each part represents the same meaning as FIGURE 6.

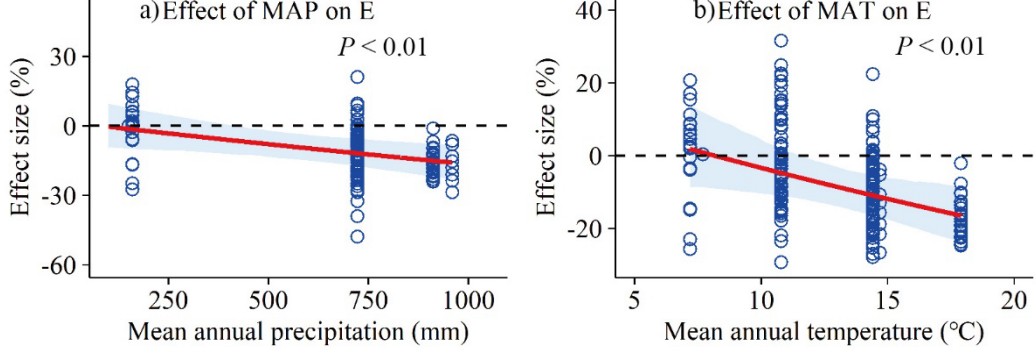

**Figure 8:** Plant mixture effects on soil evaporation in relation to mean annual precipitation (MAP) and mean annual temperature (MAT). Each part represents the same as FGURE 6.





**3.4 Effect of soil properties and management practices on water cycle in plant mixtures**

The soil properties analyzed include SOC, total nitrogen content, pH, and bulk density. For the entire dataset, SOC led to a significant decrease in the effect size of SWC, whereas bulk density increased the effect size (FIGURE 9 a,b). For croplands, the effect size of SWC increased as total nitrogen increased, whereas it decreased with SOC (FIGURE 9 c,d). Plant mixtures had no effect on other water cycle process base on soil properties or management practices.

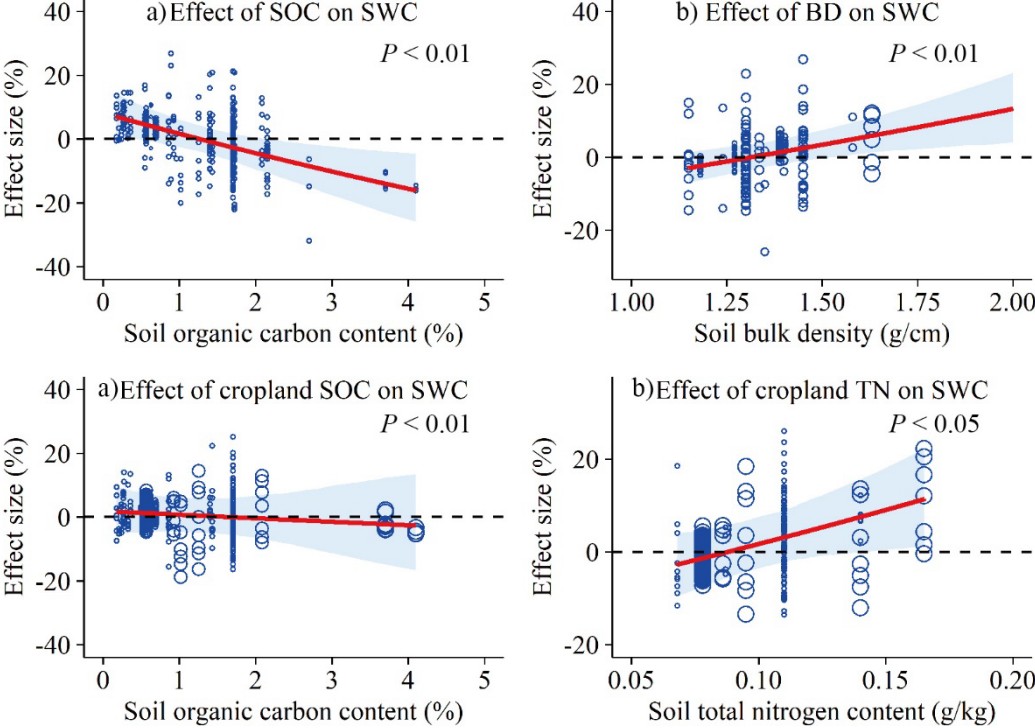

**Figure 9:** Plant mixture effects on soil water content in relation to soil properties, a) soil organic content, b) bulk density for full ecosystem types and c) soil organic content, d) soil total nitrogen for crop mixtures.

The management practices analyzed included crop type (leguminous or non-leguminous), fertilization, and irrigation pattern. When legumes were absence from grassland ecosystems, SWC decreased significantly; however, when legumes were included, SWC increased. The inclusion of legumes in an agroforestry ecosystem resulted in a significant reduction in RO, whereas their absence had the opposite effect (FIGURE 10 e). Fertilizer use resulted in a significant reduction in Th. Irrigation resulted in increased E compared to rainfed conditions (FIGURE 10 d).





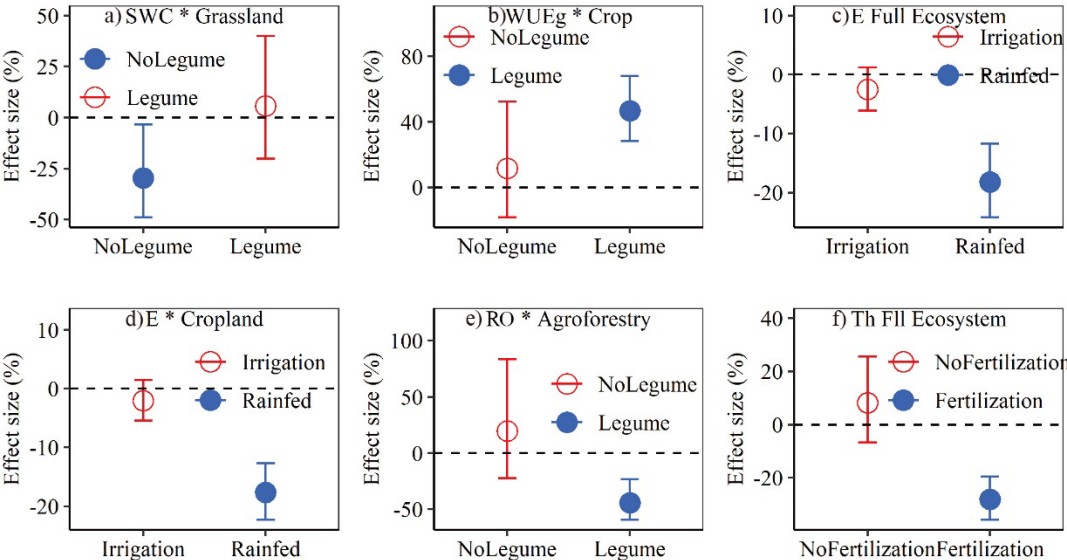

**Figure 10:** Effect size of plant mixture effects on water cycle processes in relation to manage practices.

## 4 Discussion

### 4.1 Effect of plant mixture on the water cycle and the underlying mechanism

Our study showed that plant mixtures facilitate a positive water cycle and maintain soil moisture content equivalent to that of intensive monocultures (Li et al., 2021; Schaub et al., 2020). We found that plant mixtures had a slight effect on topsoil water content, but this effect decreased with soil depth (FIGURE 3). It is possible that deep-rooted species consume more soil water in the mixture plots, despite the fact that more water entered the soil during secondary rainfall. This is consistent with the finding of Kimberly et al. (2019), who showed that plants in more diverse assemblages adjust their soil water uptake to the deeper layers (30 and 60 cm). Our results also revealed that RO was lower in plant mixtures than in monocultures, which agrees with the meta-analysis conducted in mixed forests (Gong et al., 2022). This may be because more aboveground biomass in mixed communities reduces Th and weakens rainfall kinetic energy. However, Ding et al. (2015) reported that agroforestry systems generated more RO than monocultures because the crop-seeding disturbance of the soil surface. Another influencing factor is interspecific competition, which may result in poorer plant performance in mixtures relative to monocultures (Mahaut et al., 2019). Plant mixtures showed increased WUE, which confirms the facilitation in mixed plants (FIGURE 2). Positive plant feedback and microbial facilitation promote biomass production and higher biomass in diversified plots reduces E and greatly enhances WUE through shading (Ren et al., 2019).

Except for croplands, the effect of plant mixtures on the water cycle is similar across multiple ecosystem types (FIGURE 4). Owing to the varied root characteristics of intercropped systems, WUE is optimized through broader hydrological niche (Zhang



et al., 2022). Plant stand age varied widely between ecosystem types (FIGURE 4). Forest and agroforestry systems with a long stand age may significantly affect soil structure, resulting in higher IR (Furey & Tilman, 2021; Li et al., 2021). Long-term

experiments have demonstrated that plant mixtures enhance SOC content and soil water storage capacity (Chen et al., 2020). However, Wu et al. (2020) indicated that rubber and grass mixtures might accelerate soil water drainage. This may also explain the neutral effect of plant mixture on SWC that we observed.

The effects of plant mixtures on ecosystem functioning are complex. Here we review the underlying mechanisms that systematically contribute to biodiversity and ecosystem functioning (BEF). Complementarity was the most common

explanation for the positive effect of plant mixtures (FIGURE 11) (Döring & Elsalahy, 2022). In the framework proposed by Barry et al. 2019, complementarity in BEF research include resource partitioning, biotic facilitation, and abiotic feedback. Several studies have confirmed that plant mixtures distribute resources in time and space (FIGURE 11 R1) (Barry et al., 2019; Freund et al., 2021; Guimarães-Steinicke et al., 2022). Diversified resource utilization promotes the performance of mixed communities compared with monocultures, which increases the interception of water by the canopy, and reduces throughfall.

Deep-root species not only use deeper water sources, but also promote the availability of water for adjacent species through hydraulic redistribution (Bayala & Prieto, 2020; Hafner et al., 2021; Oram et al., 2018). However, recent investigations have revealed some discrepancies. In a meta-analysis of diversified temperate grasslands, Barry et al., (2020) concluded that there was insufficient evidence to prove the existence of spatial resource partitioning across vertical gradients.

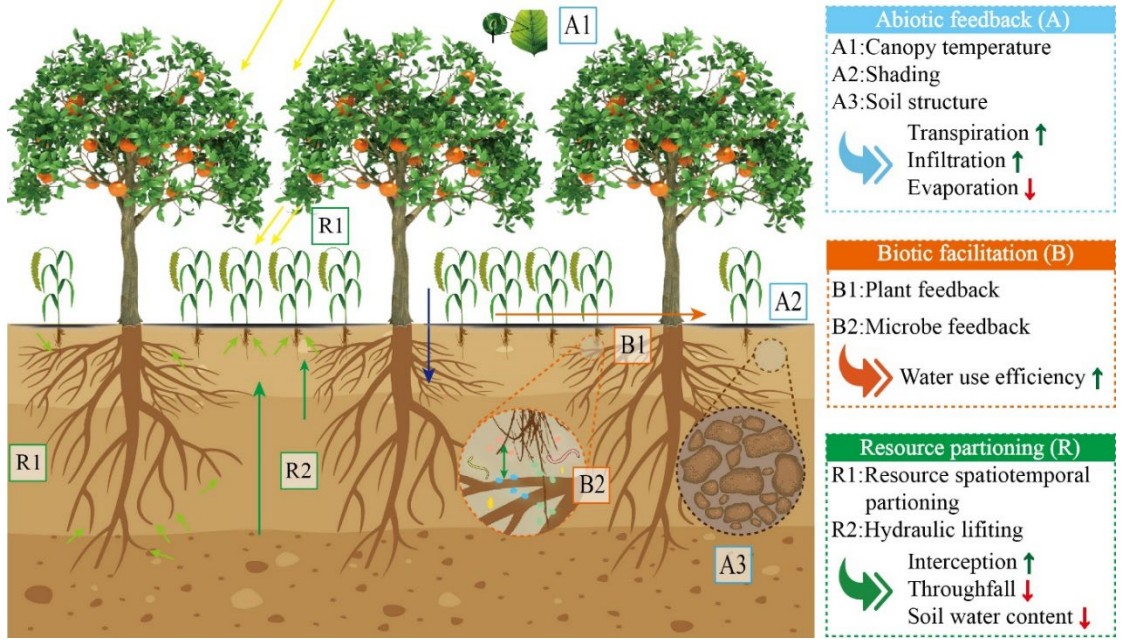

**Figure 11:** The underlying mechanisms of plant mixtures impact on water cycle processes.

Mechanisms that maintain BEF can act simultaneously. Facilitation, which via alteration of the abiotic properties of the environment by cohabiting species, essential to the maintenance of BEF, through abiotic feedback (Barry et al., 2019). We



observed that plant mixtures improved soil C, N, and P content, as well as microbial turnover rates (Prommer et al., 2020) and soil quality possibly due to the redistribution of water and nutrients by deep-rooted species (FIGURE 11 R2). A healthy water

cycle profits from suitable microclimates facilitated by plant mixtures (FIGURE 11 A1, A2) (Aguirre et al., 2021). A dense canopy, for example, helps to intercept precipitation, absorb more radiation, and reduce soil evaporation to maintain cooler vegetation surfaces and higher leaf transpiration (Guimarães Steinicke et al., 2021). Biotic facilitation from other trophic levels is also an important factor in the water cycle, although it has not been quantified in this study (FIGURE 11 B1, B2) (Schöb et al., 2018; Yu et al., 2021). Biotic feedback may enhance the performance of plant mixtures by inhibiting the establishment of

enemies and promoting the recruitment beneficial mutualists. The interaction among trophic levels could also facilitate contrasting functional performance in mixtures relative to monocultures (Homulle et al., 2022; Yu et al., 2022).

Selection effect may be another mechanism explaining the positive effects of plant mixtures (Loreau & Hector, 2001). The "selection effect" refers to the likelihood that higher function species are present. For example, shrubs may be more important than trees in reducing runoff. Thus assemblages containing a higher proportion of shrubs will have lower levels runoff than

those with less shrubs (Jie et al., 2008). Selection and complementarity effects often occur simultaneously in diverse biological systems, with their dominance being determined by season and environmental context (Zhang, Gao et al., 2021). The plant morphology of dominant species in the biodiverse communities has a significant impact on the water cycle, including canopy structure, plant height, and leaf size, which further impact throughfall (Bordoloi & Ng, 2020). When selection effects predominate, monocultures may outperform plant mixutres (Wang et al., 2021). However, positive interactions between

different species do not always exist. Species with different functional traits are more likely to promote each other's performance, where species with the same functional traits are more likely to compete with each other (Bongers et al., 2021; Guiz et al., 2018). The interactions between species are influenced by environmental factors and genetic evolution, which can promote the complexity of interactions (Schöb et al., 2018). Therefore, some mixed-species systems negatively affect on water cycle processes, such as through increased runoff.

**4.2 Environmental and management effects**

Our results showed that a higher MAP and MAT drive the effect of plant mixtures in the water cycle. Although topsoil water content (0–20 cm) decreased with increasing MAP (FIGURE 6 a), the effect size of SWC in the whole soil profile and deep soil layer (below 20 cm) was positively correlated with MAP (FIGURE 6 b-e, f). The variation in surface- and deep-SWC with MAP in plant mixtures could be due to differences in plant water use strategies. Compared with monocultures, plant mixtures

had a strong association with deep-rooted species, especially in areas with lower precipitation, which results in increased soil water uptake from deeper soil layers (Gao et al., 2018; Zhang et al., 2022). These results are consistent with those of Guderle et al. (2018), who reported that root uptake patterns were more obvious in diversified plots during periods of water scarcity. Meanwhile, the influence of MAP on SWC showed the same trend in each soil layer and changed from negative to positive with increase in MAP (FIGURE 7). Abiotic feedback (FIGURE 11), the positive effect of plant mixture on biomass in turn



influences the water cycle. The negative effect of MAP and MAT on soil evaporation was explained by the increased shading effect of increased biomass (FIGURE 8) (Guimarães Steinicke et al., 2021).

Our results showed that soil properties, including initial soil bulk density, organic carbon content, total nitrogen content and available phosphorous content, had a significant effect on SWC (FIGURE 9). The effect size decreased as SOC concentration increased, indicating that, in areas with high organic carbon content, mixed species consumed more water than single species

(FIGURE 9 a). Previous studies have shown that the positive effects of plant diversity on productivity and soil enzyme activity are more pronounced in nutrient-poor soils (Curtright & Tiemann, 2021). Plant mixtures can rapidly enhance SOC content and increase soil water storage capacity. In contrast, in nutrient-rich areas, plant mixtures have a slow enhancement of SOC content, and plant transpiration may be the main factor affecting the water cycle, resulting in lower SWC after mixing than in the corresponding monoculture (Muhammad et al., 2021). Soil structure also play a critical role in determining the soil water

storage capacity. In general, soil porosity and soil water-holding capacity decreased as soil bulk density increased. However, plant mixtures can enhance soil porosity by stimulating root growth and microbial activity, thereby increasing SWC in areas with higher bulk density areas (Ghahremani et al., 2021).

The analysis of management practices showed that legume mixtures had the most significant effect on water cycle processes (FIGURE 10 a, b, e). Legumes in mixed communities can improve nitrogen content of the deeper soil layer via nitrogen

fixation, which in turn promotes root growth in the deeper soil layers and deep soil water use (Kimberly et al., 2019). However, some studies have demonstrated that BEF relationships were not explained by increased legume abundance, but rather by the plant communities of the arid grassland's complementary use of soil water (Yu et al., 2020). Our results revealed that the absence of legumes in grassland ecosystems significantly reduced the effect size of SWC (FIGURE 10 a). This may be because the water consumption of plant mixtures was significantly lower than that of the corresponding single leguminous communities

(St Aime et al., 2020).

### 4.3 Implications and limitations

In the face of widespread biodiversity loss and land use change, plant mixtures offer great potential for the conservation of water and soil, regulation of climate, and maintenance of ecosystem services (Chen & Chen, 2021; Duffy et al., 2017; Li et al., 2021). With the global challenges of climate change and water resource scarcity, it is important than ever to understand the

ability of plants to regulate the water cycle. Overall, our results show that plant mixtures had a significant effect on multiple water cycle processes by increasing IR and WUE and decreasing RO and E. Leaves and roots affect water cycle and promote the infiltration of rainwater into the soil and their uptake by plants. These terrestrial plant systems constantly provide natural filtration and storage systems that supply a high percentage of freshwater globally. Considering all the evidence of their positive effect on multiple ecosystem services, plant mixtures could contribute to fulfilling the Sustainable Development Goals

(Blicharska et al., 2019).

Although our study disentangled the relationship between plant mixture and water cycle, we acknowledge some limitations in data collection and analysis. First, the number of observations varied among different water cycle components. The data related

to SWC accounted for the majority of observations (59.29%), whereas the data for infiltration and throughfall accounted for only 1.72% and 1.84%, respectively. Some water cycle processes, such as rainfall interception and stemflow, have received little research attention. Additionally, most studies did not report data on plant mixture relative to monocultures, i.e., only the average performance of the mixture treatments was reported; therefore the expected performance of mixtures relative to monocultures could not be accurately calculated and was excluded (Clark et al., 2019). Similarly, the standard error or standard deviation for the data used in this study was often missing. Finally, information regarding intercropping systems over longer durations (>3 years) is limited. To further elucidate the mechanism of plant mixtures on the water cycle, additional studies are needed to provide greater insight into the advantages and adaptability of plant mixtures.

## 5 Conclusions

We conducted a global meta-analysis of matched single- and multi-species plantations to evaluate the impact of plant mixtures on the water cycle. In this meta-analysis, we analyzed data from 115 studies with 1,631 paired observations from 88 publications incorporating water cycle processes, and considering the effects of climate, soil properties, and management practices. Our results demonstrate that, although the soil water storage capacity of plant mixtures was not significantly greater than that of monocultures, plant mixtures may mitigate inefficient water consumption. The topsoil water content declined slightly with annual precipitation but increased in deeper layers, demonstrating that plant mixtures may absorb more surface soil water than monocultures. We observed significant effects of SOC, crop type, and irrigation on the water cycle, particularly on soil moisture. Considering the positive effects of plant mixtures on water fluxes, it may be an effective way to simultaneously counteract biodiversity loss and climate change. However, more systematic studies are urgently needed to reveal the effects of plant mixtures on complete water cycle fluxes, especially rainfall interception, stemflow, and evapotranspiration, specifically with regards to long-term studies.

**Data availability**

The data used in this study are available on Dryad Digital Repository (https://doi.org/10.5061/dryad.j6q573njf).

**Author contributions**

XN Zhao and XD Gao conceived the study; HQ Liu collected the data and performed the analysis and prepared the first draft of the manuscript with the help of CJ Li; YH Cai and XL Song edited and commented on the manuscript.

**Competing interests**

The authors declare no conflicts of interest relevant to this study.



**Acknowledgments**

This work was jointly supported by the National Key Research and Development Program of China (2021YFD1900700), the Cyrus Tang Foundation, the Shaanxi Key Research and Development Program (2020ZDLNY07-04).

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
