# Peer review of "Global meta-analysis of water cycling response to plant mixture"

_EGUsphere, 2023_

## Author Comment (AC1)

**General comments**

This meta-analysis work analyzed how does plant mixture affect land surface water cycle. This topic is very important and this work is timely to inform what is the situation. The authors collected a very good data, and the methodological framework is robust. Therefore, this is a nice and solid work. I'm glad to read it at this stage. I have no big major concern except several technical points.

Response: Thank you very much for your time and constructive comments on our paper. We have read through comments carefully and have made modifications. Please see the following point-by-point replies for details.

Line 28: I would like to remove "global", as plants play their roles mainly in the local - regional scales, and their effects are largely limited in the global scale.

Response: Thank you for your comment. This sentence will be rewritten in the text (Line 28).

"Plants are the main biological regulators of regional water cycles and determine the hydrological fluxes of ecosystems."

Line 38: remove the dot

Response: It has been edited in the text (Line 38).

Line 40: what does the "this" mean is unclear, rephrase please

Response: Thank you for your comment. The term "this" in Line 40 refers to substrate change. This sentence will be rewritten in the text as follows (Line 40-42).

"These modifications in the substrate affect the distribution of precipitation between the above- and below-ground components subsequently impacting a broad spectrum of processes within the water cycle, such as evaporation, stemflow, throughfall, soil infiltration, and surface runoff (Collins & Bras, 2007; Luo et al., 2022; Li et al., 2022; Unigarro et al., 2023)."

Line 50-51: it's unclear what is "the region"? clarify

Response: Thank you for your comment. This sentence will be rewritten in the text as follows (Line 51-52).

"However, various studies have reported contradictory results (Rahman et al., 2017; Guderle et al., 2018) or negligible (O'Keefe et al., 2019) effects of plant mixtures on soil water content as well as for other water cycle processes (Ghahremani et al., 2021; Leimer et al., 2014; Spehn et al., 2000)."

Line 80-82: it will be better to write the search expression in a formal way, which will benefit reproduction

Response: Thank you for your comment. We will add search expression as follows in the text (Line 82-84).

"We identified peer-reviewed papers published up to 1 Jun 2022 in the Web of science, SCOPUS, and China Knowledge Resource databases with the search term 'biodiversity OR "plant diversity" OR "species richness" OR "species diversity" OR "species abundance" OR monoculture OR mixture AND "water fluxes" OR "water cycle" OR runoff OR infiltration OR evaporation OR evapotranspiration OR transpiration OR "soil water" OR interflow OR throughfall."

Line 103: please provide details about the data source

Response: We agree. We will add search data source as follows in the text (Line 108).

"Ecosystem types were categorized as forests, croplands, agroforest, grasslands, and indoor plantation (Chen & Chen, 2021)."

References:

Chen, X., & Chen, H. Y. H. (2021). Plant mixture balances terrestrial ecosystem C:N:P stoichiometry. *Nature Communications*, 12(1). doi:10.1038/s41467-021-24889-w

Line 167: replace these abbreviation names with "water cycle" or similar words, to make it simpler and clearer

Response: We agree. This sentence will be rewritten in the text as follows (Line 172-173).

"**Figure 4:** Comparison of water cycles (Soil water content, runoff, steady infiltration, evaporation, transpiration, throughfall, and water use efficiency) in diversification assemblages versus monocultures between ecosystem types."

Line 188: "at" → "among"

Response: We agree. It will be revised in the text.

Line 189: in the topsoil (0-20 cm) in deeper soil? An "and" is missing here?

Response: Thanks. This sentence will be rewritten in the text as follows (Line 195).

"The effect of plant mixture on SWC exhibited a slight increase in correlation with MAP throughout the entire soil profile."

Figure 10: legend within each panel is not necessary. It's better to remove them to keep the figure more concise

Response: We agree. It will be edited in the revised manuscript.

---

## Author Comment (AC2)

**General comments**

This paper investigates the effect of plant mixture on water cycle process. The authors perform a meta-analysis, using 1631 paired observations (monocultures-mixtures) from 88 published studies. They conclude that, compared to monoculture, plant species mixtures can promote positive hydrological processes by improving plant water use and reducing unproductive water consumption. Given the relatively fewer information on hydrological processes in relation to plant mixture in previous studies, compared to biomass and soil properties, the study is of great interest.

However, the descriptions of statistical analyses are incomplete, and the explanations of how to analyze the effects of mean annual precipitation and temperature on the relationship between plant mixtures and water cycling, as well as the roles of other influencing factors, are unclear.

In the discussion, the mechanisms underlying the impact of plant mixtures on hydrological processes are mainly attributed to complementarity and interspecific facilitation, with selection effects and complementarity effects also mentioned. These theories form the foundation for understanding the relationship between biodiversity and ecosystem functions. However, the article does not effectively integrate these theories with the research results.

Response: Thank you very much for your time and constructive comments on our paper. We have read through comments carefully and have made modifications. Please see the following point-by-point replies for details.

However, the descriptions of statistical analyses are incomplete, and the explanations of how to analyze the effects of mean annual precipitation and temperature on the relationship between plant mixtures and water cycling, as well as the roles of other influencing factors, are unclear.

Response: Thank you for your comment. We will add descriptions of statistical analyses as follows in the text (Line 135-142).

"To analysis the impact of MAT and MAP on the changes in water cycling caused by plant mixtures, we incorporated terms for MAT and MAP into Equation 3 (Peng & Chen, 2021). The optimal model was selected based on the AIC values. Ultimately, this allowed us to determine the variations in the effects of MAT and MAP on plant mixtures at different soil depths.

To examine whether the species mixture effects changed with influencing factors— such as ecosystem type, management practices, seeding pattern, fertilizer use or plant type—we conducted an analysis using the linear mixed-effects model as follows:

$$\ln RR = \beta_0 + \beta_1 \times F + \pi_{study} + \varepsilon \tag{4}$$

where $\beta_s$ are the coefficients to be estimated, $F$ is the influencing factor, other implications as previously mentioned, $\pi_{study}$ is the random effect factor of the study, and $\varepsilon$ is the sampling error."

In the discussion, the mechanisms underlying the impact of plant mixtures on hydrological processes are mainly attributed to complementarity and interspecific

facilitation, with selection effects and complementarity effects also mentioned. These theories form the foundation for understanding the relationship between biodiversity and ecosystem functions. However, the article does not effectively integrate these theories with the research results.

Response: Thanks for your time and constructive comments. We will reorganize section "4.1 Effect of plant mixture on the water cycle and the underlying mechanism" as follows in the text (Line 225-275).

"The effects of plant mixtures on ecosystem functioning are multifaceted. Here we review the underlying mechanisms systematically that contribute to biodiversity and ecosystem functioning (BEF), which a particular focus on studies related to the water cycle. Our results demonstrated the beneficial effects of plant mixtures on the water cycle, as it can maintain soil moisture content comparable to that of monocultures while increasing productive water consumption (Li et al., 2021; Schaub et al., 2020). Complementarity was the most prevalent explanation for the positive effect of plant mixtures (Figure 11) (Döring & Elsalahy, 2022; Barry et al., 2019). Numerous studies have confirmed that temporal and spatial niche differentiation among various species facilitates resource partitioning (Figure 11 R1) (Barry et al., 2019; Freund et al., 2021; Guimarães-Steinicke et al., 2022). We observed that plant mixtures had a slight effect on the whole soil profile. However, with increasing soil depth, these mixtures consume more water than monocultures (Figure 3). Deep-rooted species may deplete more soil water in the mixture plots, despite the fact that more water entered the soil during rainfall events (Lange et al. 2019). This is consistent with the finding of Kimberly et al. (2019), who showed that plants in more diverse assemblages adjust their soil water uptake to the deeper layers (30 and 60 cm). Plant mixtures increase soil water consumption and occur more frequently in areas with lower annual rainfall (Figure 6). Insufficient surface water forces deep-rooted species in mixed communities to alter their root morphology and absorb more deep soil water (Zhao et al. 2023). The impact of plant mixtures on soil water is a comprehensive reflection of multiple water cycling processes, such as increase canopy interception of water, reduce rainfall kinetic energy, and consequently reduce throughflow and runoff (Figure 2). Additionally, they may enhance soil water-holding capacity by changing soil properties, allowing more water to enter the soil during rainfall events (Gong et al., 2022; Lange et al., 2019).

Facilitation, which occurs through the alteration of the abiotic properties of the environment by cohabiting species, is crucial to the maintenance of BEF, through abiotic feedback (Barry et al., 2019). Deep-root species not only use deeper water sources, but also promote the availability of water for adjacent species through hydraulic redistribution (Figure 11 R2) (Bayala & Prieto, 2020; Hafner et al., 2021; Oram et al., 2018). Plant mixtures showed increased WUE, infiltration rate and decreased evaporation (Figure 2), which confirms the positive plant and microclimate facilitation in mixed plants (Ren et al., 2019). This abiotic factor facilitation may be a crucial factor in improving soil carbon, nitrogen, and phosphorus content, as well as microbial turnover rates (Prommer et al., 2020). Suitable microclimates resulting from mixed plantations also promote a healthy water cycle (Figure 11 A1, A2) (Aguirre et al., 2021). A dense canopy, for example, absorbs more radiation and enhances plant

transpiration, maintaining lower vegetation surface temperature, and reducing the inhibitory effect of high temperatures on transpiration (Guimarães Steinicke et al., 2021).

Biotic facilitation from other trophic levels is also an important factor in the water cycle, although it has not been quantified in this study (Figure 11 B1, B2) (Schöb et al., 2018; Yu et al., 2021). Biotic feedback may enhance the performance of plant mixtures by inhibiting the establishment of enemies and promoting the recruitment beneficial mutualists. The interaction among trophic levels could also facilitate contrasting functional performance in mixtures relative to monocultures (Homulle et al., 2022; Yu et al., 2022).

Selection effect may be another mechanism explaining the positive effects of plant mixtures (Loreau & Hector, 2001). The "selection effect" refers to the likelihood that higher function species are present. For example, shrubs may be more important than trees in reducing runoff. Thus, assemblages containing a higher proportion of shrubs will have lower levels runoff than those with less shrubs (Jie et al., 2008). Selection and complementarity effects often occur simultaneously in diverse biological systems, with their dominance being determined by season and environmental context (Zhang, Gao et al., 2021). Positive interactions between different species do not always exist. When selection effects predominate, monocultures may outperform plant mixtures (Wang et al., 2021). The plant morphology of dominant species in the biodiverse communities has a significant impact on the water cycle, including canopy structure, plant height, and leaf size, which further impact throughfall (Bordoloi & Ng, 2020). Species with different functional traits are more likely to promote each other's performance, where species with the same functional traits are more likely to compete with each other (Bongers et al., 2021; Guiz et al., 2018). For instance, Ding et al. (2015) reported that agroforestry systems generated more runoff than monocultures because the crop-seeding disturbance of the soil surface. Another important influencing factor is interspecific competition, which may result in poorer plant performance in mixtures relative to monocultures (Mahaut et al., 2019). Furthermore, species interactions are influenced by environmental factors and genetic evolution, which can promote the complexity of interactions (Schöb et al., 2018)."

---

## Author Comment (AC3)

This meta-analysis examines the effects of plant mixtures on various processes of the water cycle and how these effects differ among ecosystem types, climate conditions, soil properties, and management practices. The authors report that plant mixtures have both positive and negative impacts, depending on the water cycle process considered and that the effect size of plant mixtures strongly depend on ecosystem types considered as well as soil and climatic factors.

**Specific comments:**

While the topic of the study is highly interesting, I have some major concerns:

The distribution of the vegetation types and also of the water cycle processes considered is highly unbalanced. The purpose of a global meta-analysis becomes questionable when there are fewer than 10 grassland studies included and for some processes, fewer than 10 publications are available (as noted in my specific comment). The limited number of publications makes it challenging to derive meaningful results, especially when comparing processes across different ecosystem types. I recommend either expanding the number of studies in the meta-analysis or focusing on ecosystems with a substantial number of studies (such as forests, agroforestry, and croplands). If the latter option is chosen, please update the title accordingly.

**Response:** Thanks a lot for your comments and suggestion. Here, we choose to expand the number of studies in the meta-analysis. First, we modify the inclusion criteria. In the original version, we excluded studies that did not provide explicit planting durations. At this time, we find that a lack of planting duration does not significantly skew our overall analysis. Therefore, we now include articles without explicit planting durations in the updated database. Second, we extend the ending time from January 1 2022 to October 1 2023 in literature survey. In this way, the whole dataset for meta-analysis is boosted. Finally, we reviewed the whole literature and added more data that were previously left out. Now, a total of 161 studies with 2,973 paired observations from 130 publications were included in the database (113 studies, 1631 paired observations and 88 publications in original database). Despite the updated database remains imbalance, the number of studies for each vegetation types is greatly increased. Moreover, we will report the number of literature and observations in the revised manuscript.

It is not clear what is a monoculture in agroforestry. "Agroforestry refers to any of a broad range of land use practices where pasture or crops are integrated with trees and shrubs [https://en.wikipedia.org/wiki/Agroforestry]. In forests it is likely the dominating tree species in a plantation. However, this needs to be defined/explained. Thus, further explanations and definitions are needed, and probably the authors need to re-evaluate the studies.

**Response:** Thanks for your comments. We have re-evaluated the article included based on this criterion. The studies in the articles by Sun et al. (2014), Li et al. (2020), and Wang et al. (2011) are about the mixture of herbaceous plants and crops. We will classify them as crops in the revised paper.

**Reference**
Sun, B., Peng, Y., Yang, H., Li, Z., Gao, Y., Wang, C., Yan, Y., & Liu, Y. (2014). Alfalfa (Medicago sativa L.)/Maize (Zea mays L.) Intercropping Provides a Feasible Way to Improve Yield and Economic

Incomes in Farming and Pastoral Areas of Northeast China. *PLoS ONE*, 9(10), e110556. https://doi.org/10.1371/journal.pone.0110556

Li E., Mu Y., He Y., Zhang X., & Yang S. (2020). Effects of wheat/alfalfa intercropping systems on soil moisture and water utilization efficiency. *Research of Soil and Water Conservation*, 27(1), 54-58+65. https://doi.org/10.13869/j.cnki.rswc.2020.01.008

Wang, L., Zhong, C., Gao, P., Xi, W., & Zhang, S. (2015). Soil infiltration characteristics in agroforestry systems and their relationships with the temporal distribution of rainfall on the loess plateau in China. *PLoS ONE*, 10(4), e0124767. https://doi.org/10.1371/journal.pone.0124767

In addition, the text needs linguistic revision. In particular, the use of scientific terms and the presentation of graphics are sometimes a bit sloppy. I have listed some, but not all, typos and incorrectly used terms below.

**Response:** Thanks for your comments. We have carefully checked the language of the manuscript with the help of a native English-speaking editor. The English will be clearly improved in the revised manuscript.

Ls13-14: Please rephrase. Since meta analyses are based on many publications, some understanding how plant mixtures impact the water cycle is already there. But, are there contrasting findings, or could findings be generalized, even over different vegetation types?

**Response:** We agree. We will change it to "However, the effects of plant mixing on the water cycle is equivocal" in the revised manuscript.

L17: In terms of functions such as transpiration "increasing" is likely more appropriated that "improved"

**Response:** We agree. We will edit the text in the revised manuscript.

L22: Do you mean "mixtures" instead of "minutes"?

**Response:** Yes. We will edit the text in the revised manuscript.

L23: "impact" or similar is more appropriated than "regulate"

**Response:** We agree. We will edit the text in the revised manuscript.

L24: What is a "positive water cycle"? please rephrase.

**Response:** Here we meant that plant mixtures could reduce soil evaporation and increase transpiration and water use efficiency. We will change it to "This work highlights the importance of plant mixture in facilitating infiltration and plant water use and provide insights into the establishment of sustainable ecosystems" in the revised manuscript.

L35: What is meant by "Plant mixtures […] not only promote species recruitment"? Maybe remove as not related to the study.

**Response:** We agree. We will edit the text in the revised manuscript.

L38: Remove dot before the references.

**Response:** We agree. We will edit the text in the revised manuscript.

Ls42-45: Unclear, what is meant by negative effects, if the result of higher diversity is improved soil water availability. Please elaborate on this.

**Response:** Thanks for your comment. The term "negative effect" is ambiguous and not suitable here. So, we will delete the sentence "Although extensive research has demonstrated the potential of plant diversity to benefit the water cycle, potential negative effects have also been highlighted" in the revised manuscript. And the following text will be edited as follows.

"For example, some studies demonstrated that overyielding of mixture relative to monoculture reduced throughfall but increased stemflow, infiltration and soil water availability (Göransson et al., 2016; Leimer et al., 2018)"

**Reference**

Göransson, H., Bambrick, M. T., & Godbold, D. L. (2016). Overyielding of temperate deciduous tree mixtures is maintained under throughfall reduction. *Plant and Soil*, *408*(1-2), 285-298. doi:10.1007/s11104-016-2930-1

Leimer, S., Bischoff, S., Boch, S., Busch, V., Escher, P., Fischer, M., ... Wilcke, W. (2018). Does plant diversity affect the water balance of established grassland systems? *Ecohydrology*, *11*(4), doi:e1945. 10.1002/eco.1945

L44: Strange wording "overproduction of mixtures". Do you mean overyielding?

**Response:** We agree. We will edit the text in the revised manuscript.

L48: Table 1 does not show that "most studies have focused only on the effects of plant mixtures on individual or several of these water cycle processes".    Please correct.

**Response:** Thanks for your comment. It is the supplementary Table S1 that lists the publications used in the meta-analysis. We will correct it in the revised manuscript.

Ls50-52: It would be beneficial if some specific examples of regional differences in the relationship between plant diversity and water cycle processes were presented in the introduction.

**Response:** We agree. The following examples will be added in the revised manuscript.

"However, existing studies have reported contradictory results or negligible effects of plant mixtures on soil water content as well as for other water cycle processes (Table 1).    For example, Rahman et al. (2017) reported that average SWC in the intercropping was relatively greater compared to sole cropping. However, Gong et al. (2020) found that the average SWC in the intercropping was lower than corresponding monoculture in the 0-50 cm soil layer, while in the 50-100 cm soil layer, the SWC in the intercropping was higher. Regarding surface runoff processes, Fan et al. (2016) found that the runoff rate for intercropping corn and potatoes were smaller compared to those of monoculture. However, Machiwal

et al. (2021) shown that runoff rate from intercropping sorghum and cluster-bean may be greater than that from monoculture of cluster-bean."

**Reference**

Gong, X., Dang, K., Lv, S., Zhao, G., Tian, L., Luo, Y., & Feng, B. (2020). Interspecific root interactions and water-use efficiency of intercropped proso millet and mung bean. *European Journal of Agronomy*, *115*(126034). doi:10.1016/j.eja.2020.126034

Fan, Z., An, T., Wu, K., Zhou, F., Zi, S., Yang, Y., ... Wu, B. (2016). Effects of intercropping of maize and potato on sloping land on the water balance and surface runoff. *Agricultural Water Management*, *166*, 9-16. doi:10.1016/j.agwat.2015.12.006

Machiwal, D., Kumar, S., Islam, A., Kumar, S., Jat, S. R., Vaishnav, M., & Dayal, D. (2021). Evaluating effect of cover crops on runoff, soil loss and soil nutrients in an Indian arid region. *Communications in Soil Science and Plant Analysis*, *52*(14), 1669-1688. doi:10.1080/00103624.2021.1892726

Rahman, T., Liu, X., Hussain, S., Ahmed, S., Chen, G., Yang, F., ... Yang, W. (2017). Water use efficiency and evapotranspiration in maize-soybean relay strip intercrop systems as affected by planting geometries. *PloS ONE*, *12*(e01783326). doi:10.1371/journal.pone.0178332

L63 (and L 235): Misleading citation, Lange et al. 2015 and Chen et al. 2020 reported on increases soil organic matter content with plant diversity. Effects on soil water holding capacity are not assessed/discussed.

**Response:** Thanks for your comment. This part will be rewritten as follows in the revised manuscript to avoid misunderstanding.

"Although no studies have yet been conducted to determine the causes of varying effects of plant mixtures on water cycle processes, research on the relationship between plant diversity and ecosystem function suggests that this variability may depend on the type of plant mixture, planting period, soil characteristics, climate, and management practices (Augusto & Boča, 2022; Feng et al., 2022; Freschet et al., 2017; Mori et al., 2020; Toïgo et al., 2021; Wright et al., 2017). For instance, Cheng et al. (2023) found that plant mixture on soil water content was species specific. Similarly, Ye et al. (2022) argued that inclusion of leguminous species significantly affect soil organic carbon content and quality"


Ls63-65: I disagree with this sentence. I think there is a good in-depth understanding of how plant diversity impacts biological and physical ecosystem properties, which in turn affect the water cycle. A global meta-study can increase the knowledge of general patterns and drivers among regions and vegetation types as well as differences among them.

**Response:** Thanks for your comment. This part will be rewritten as follows in the revised manuscript.

"Despite extensive research on how plant diversity affects water cycle processes, there is currently no global consensus on the overall impact of plant diversity on hydrological processes. Furthermore, plot-scale studies make it difficult for us to understand the impacts of different ecosystem types, soil characteristics, and management measures."

Ls86-92: Based on Figure 1, 4 and Table S1, the ecosystem types and the water cycle processes are very unbalanced in this meta analyses. For instance, only a few grasslands are included in this study and just a few studies that investigate soil evaporation (7), throughfall (3), leaf transpiration (7). Though, it is possible to calculate effect sizes based on only a few studies/observations, I think this is a caveat for a general global meta-analysis.

**Response:** Thank you for your comment. In order to address this issue, we have included the recently published article and modify the search strategy, resulting in the inclusion of a greater number of grassland ecosystems in our analysis. The database now includes 24 grassland-related studies (an increase of 14). Among all the studies, 10, 4, and 13 studies respectively for soil evaporation, throughfall, and leaf transpiration, all of which have clearly increased compared with initial data. Here, we use the "fixed effects model" to calculate studies with fewer than three items, and clearly mark the number of studies and observations used in different analyses (Figure 4). All of these revisions will be reported in the revised manuscript.

[Figure]

**FIGURE 4:** Water cycles in plant mixtures versus monocultures between ecosystem types. Means and horizontal error bars represent means and 95% confidence intervals (CIs) for plant mixture effects. The numbers outside the parentheses indicated the number of observations pairs, while inside the parentheses indicated the number of publications. SWC, RO, E, IR, LT, Th and WUE represent soil water content, runoff rate, evaporation, transpiration, throughfall and water use efficiency, respectively.

L99: "seed" or "species" sown?

**Response:** Thanks for your comment. Within grassland ecosystems, we calculate the proportion of each species in the mixture treatment based on the seed ratio.

L99: Please elaborate on the rationale and methods of the simulations.

**Response:** In this context, "Simulation" refers to indoor simulation experiments, including pot or soil column experiments. In the revised manuscript, 10 indoor experiments will be included for analysis, and "simulation" ecosystem type will be classified into crops (Ouyang et al., 2018, Wan et al., 2021, Zhu et al., 2023, Zhou et al., 2023) and grasslands (Nagase & Dunnett, 2012, Xu et al, 2022, Liu et al., 2021, Su et al., 2019, Zhang et al., 2017, Xu et al., 2023).


L106: What does "available" mean, "plant available"? Maybe use "mineral" instead.

**Response:** We agree. We will edit the text in the revised manuscript.

L142: Do you mean "maximum likelihood" instead of "restricted likelihood".

**Response:** We agree. We will edit the text in the revised manuscript.

L162: This seems a bit selective. Based on Figure 4, only water use efficiency was different in croplands from grassland, but see my earlier comment the unbalanced number of studies for ecosystems.

**Response:** Thanks, we have revised it as "The effect size of plant mixtures on the water cycle was similar between ecosystem types, except for runoff process in croplands (FIGURE 4).".

[Figure]

**FIGURE 4:** Water cycles in plant mixtures versus monocultures between ecosystem types. Means and horizontal error bars represent means and 95% confidence intervals (CIs) for plant mixture effects. The numbers outside the parentheses indicated the number of observations pairs, while inside the parentheses indicated the number of publications. SWC, RO, E, IR, LT, Th and WUE represent soil water content, runoff rate, evaporation, transpiration, throughfall and water use efficiency, respectively.

Ls163-164: Unclear. Please rephrase.

**Response:** We have revised it as "In contrast to forests and agroforestry ecosystems, crops have a shorter planting period (FIGURE 3). Furthermore, within the database we incorporated, the proportion of crop studies is notably high, accounting for 92 out of a total of 161 studies."

L189: Unclear. Please rephrase.

**Response:** Based on the updated results, we have revised it as "The effect of plant mixture on SWC increased slightly with MAP in the whole soil layer"

Ls189-190: Unclear. Please rephrase. What exactly does the significant interaction term between MAT and sampling depth tell; different directions, different slopes?

**Response:** We have revised it as "MAT had a significant effect on SWC with soil depth (MAT×SD, P < 0.001) (FIGURE 7). The significant interaction indicates that the impact of MAT on the effect size varies with soil depth."

L211: Not clear what kind of plant communities were compared when assessing the effect of legumes on the diversity effect: non-legume monocultures vs non-legume mixtures, legume monocultures vs legume mixtures, non-legume monocultures vs. legume mixtures,… This should be defined.

**Response:** Thanks for your comments. The impact of hybrid legume crops is not compared in a single study, but rather each effect value is categorized into two types: one with legume plants (Y) and the other without legume plants (N). We analyze the impact of legume plants by comparing studies that include legume plants with those that do not.

L239: Delete "and".

**Response:** Thanks for your comments. The phrase "biodiversity and ecosystem functioning" is commonly used in the study of biodiversity and ecosystem functions, as shown in the literature below.


Figures and Tables: Please provide information on the abbreviations in all figures (e.g. missing in Figure 2 and Figure S3, Table S1), so that all figures can be understood on a stand-alone basis.

**Response:** We agree. We will give the full name of all abbreviations in the figure caption for all figures in the revised manuscript.

Figure 2: Please rephrase the captions, unclear.

**Response:** We agree. The caption of the figure will be edited as follows in the revised manuscript.

"Figure 2: Effect size of plant mixtures on water cycle processes versus monocultures. Solid circles mean significant effect ($P < 0.05$), hollow circles mean no significant effect. The numbers outside the

parentheses indicated the number of observations pairs, while inside the parentheses indicated the number of studies. SWC, RO, E, IR, LT, Th and WUE represent soil water content, runoff rate, evaporation, transpiration, throughfall and water use efficiency, respectively."

Figure 3b: I am wondering if the trendline would be similar, if the single observation at 15 years is not included. The huge confidence interval is also remarkable.

**Response:** In FIGURE 3, data points with longer planting years generally attributes to forest ecosystems. Analyzing by different ecosystems can effectively address the impact of planting years in different ecosystems on the overall results (FIGURE S3). As you mentioned, due to data imbalance, there is a significant discrepancy between the overall results in Figure 3 and the results of different ecosystem types. Some trends, such as throughfall (FIGURE 3E), cannot be attributed to the influence of planting years. After modifying according to your suggestion, it can better reflect the actual situation.

[Figure]

**FIGURE 3** Plant mixture effect on the water cycle processes in terms of plant stand age and soil moisture measurement depth. A-G shown plant stand age, H shown soil depth. SWC, RO, E, IR, LT, Th and WUE represent soil water content, runoff rate, evaporation, transpiration, throughfall and water use efficiency, respectively. The black lines are fitted effect sizes, with bootstrapped 95% confidence intervals shaded in grey. The size of circles (Wr) represents the relative weights of corresponding observations.

[Figure]

**FIGURE S4** Effect of plant mixture on water cycle processes in terms of stand age. A-E denote crops, F-H denote forests, I-J denote grasslands, and K denote agroforestry, respectively. SWC, RO, E, IR, LT, Th and WUE represent soil water content, runoff rate, evaporation, transpiration, throughfall and water use efficiency, respectively. The black lines are fitted effect sizes, with bootstrapped 95% confidence intervals shaded in grey. The size of circles represents the relative weights of corresponding observations. Figures 5 & 6: What is the "relative weight", how and why was it calculated? Also, the red lines in most panels do not properly represent the distribution of points (observations), in particular in the panels 100-200mm and 200-500mm. Please explain.

**Response:** "The relative weight" refers to the weight we assign to each study when conducting our analysis. Here, the number of replications was used to calculate the relative weight (Pittelkow et al., 2015).

$$Wr = \frac{(N_c \times N_t)}{(N_c + N_t)} \tag{1}$$

where $Wr$ is the weight for observed values and $Nt$ and $Nc$ are the number of replications in mixtures and monocultures, respectively.

The following model was employed to test the effects of MAP, MAT and soil depth (SD) on the lnRR of SWC, RO, IR, E, Th, LT, and WUE:

$$\ln RR = \beta_0 + \beta_1 \times MAP + \beta_2 \times SD + \beta_3 \times MAP \times SD + \pi_{study} + \varepsilon \qquad (2)$$

$$\ln RR = \beta_0 + \beta_1 \times MAT + \beta_2 \times SD + \beta_3 \times MAT \times SD + \pi_{study} + \varepsilon \qquad (3)$$

where $\beta_s$ are the coefficients to be estimated; $MAP$ is mean annual precipitation; $MAT$ is mean annual temperature; $\pi_{study}$ is the random effect factor of the study, and $\varepsilon$ is the sampling error.

In our analysis, we took into account the interaction between MAP or MAT and soil depth, as indicated by MAP×SD and MAT×SD in equations 2 and 3. The regression slopes for different soil layers are influenced not only by the data points of the current layer but also by the results of other layers. Our findings indicate significant interaction effects of 0.018 (p<0.001) for MAP×SD and 0.017 (p<0.001) for MAT×SD. As a result, the regression slopes for the 100-200cm and 200-500cm soil depths display inconsistencies with the data points. This discrepancy also appeared in our analysis results after updating our data, and we conducted a grouped regression analysis, as depicted in the subsequent figure.

[Figure]

**FIGURE 6:** Interactive effects of mean annual precipitation (MAP) and soil moisture measurement depth (SD) on the effect size of plant mixture on soil water content. The black line represents the estimated mean response, with bootstrapped 95% confidence intervals shaded in blue. The figure was plotted based on the most parsimonious models derived from Equation 7. The size of circles (Wr) represents the relative weights of corresponding observations.

[Figure]

**FIGURE 7:** Interactive effects of mean annual temperature (MAT) and soil depth (SD) on the effect size of plant mixture on soil water content. The **black** line represents the estimated mean response, with bootstrapped 95% confidence intervals shaded in blue. The figure was plotted based on the most parsimonious models derived from Equation 7. The size of circles (Wr) represents the relative weights of corresponding observations.

[Figure]

**FIGURE** Effects of MAP on the effect size of plant mixture on SWC. The blue line represents the estimated mean response, with bootstrapped 95% confidence intervals shaded in grey. The figure was plotted based on the liner models derived from different SD subgroup. The size of circles (Wr) represents the relative weights of corresponding observations.

[Figure]

**FIGURE** Effects of MAT on the effect size of plant mixture on SWC. The blue line represents the estimated mean response, with bootstrapped 95% confidence intervals shaded in grey. The figure was plotted based on the liner models derived from different SD subgroup. The size of circles (Wr) represents the relative weights of corresponding observations.

**Reference**

Pittelkow, C. M., Liang, X., Linquist, B. A., van Groenigen, K. J., Lee, J., Lundy, M. E., ... van Kessel, C. (2015). Productivity limits and potentials of the principles of conservation agriculture. *Nature*, 517(7534), 365-368. doi:10.1038/nature13809

Figure 9: Check letters on the panels. In the captions, add carbon to "soil organic content".

**Response:** Thanks for your suggestion. We will edit this figure in the revised manuscript.

[Figure]

**Figure S5** Plant mixture effects on water cycle processes in relation to influencing factors, A, SWC related to SOC; B, SWC related to pH; C, E related to MAP; D, E related to MAT; E, E related to SOC; F, E related to TN; G, IR related to SOC; H, IR related to Clay; I, IR related to pH; J, RO related to SOC; K, RO related to TN; L, RO related to Sand; M, RO related to Silt; N, Th related to Clay; O, Th related to Silt; P, LT related to SOC; Q, LT related to Sand; R, LT related to Clay; S, LT related Silt; T, WUE related to TN; U, WUE related to pH; V, WUE related to BD. SWC, RO, E, IR, LT, Th and WUE represent soil water content, runoff rate, evaporation, transpiration, throughfall and water use efficiency, respectively.

Table S1: Typo in the captions: should be "throughfall". Also, three of the 88 studies do not have any entry in the responses of water cycle processes to plant mixtures? In addition, I strongly suggest to include information on ecosystem types, the studies deal with.

**Response:** Thanks. We will complete the missing information in Table S1, and add information on ecosystem types in Table S1.

**Table S1** Reviewed references of the responses of water cycle processes to plant mixtures in this meta-analysis.

| Reference ID | Reference | Ecosystem type | SWC | IR | RO | E | LT | Th | WUE |
|---|---|---|---|---|---|---|---|---|---|
| 1 | Altinalmazis et al., 2020 | Forest | Yes | | | | | | |
| 2 | An et al., 2014 | Crop | | | Yes | | | | |
| 3 | An et al., 2019 | Crop | | | Yes | | | | |
| 4 | Ashilenje et al., 2023 | Crop | Yes | | | | | | |
| 5 | Chai et al., 2011 | Crop | | | | Yes | | | |
| 6 | Chen & Zheng, 2018 | Crop | | | | Yes | | | |
| 7 | Chen et al., 2008 | Forest | Yes | | Yes | | Yes | | |
| 8 | Chen X et al., 2015 | Crop | | | Yes | | | | |
| 9 | Chen G et al., 2015 | Crop | | | | | | | Yes |
| 10 | Chen et al., 2016 | Forest | | | | | Yes | | |
| 11 | Chen et al., 2020 | Forest | | Yes | | | | | |
| 12 | Cheng et al., 2022 | Forest | Yes | | | | | | |
| 13 | Chimonyo et al., 2016 | Crop | | | | | | | Yes |
| 14 | Chirwa et al., 2003 | Forest | | Yes | | | | | |
| 15 | Collins et al., 2017 | Agroforestry | Yes | | | | | | |
| 16 | Ding et al., 2015 | Agroforestry | | | Yes | | | Yes | |
| 17 | Du et al., 2017 | Crop | | | Yes | | | | |
| 18 | Fan & Wu et al., 2016 | Crop | Yes | | | | | | |

| | | | | | | | | |
|---|---|---|---|---|---|---|---|---|
| 19 | Fan et al., 2016 | Crop | Yes | | Yes | Yes | | |
| 20 | Fang et al., 2020 | Forest | | Yes | | | | |
| 21 | Forrester et al., 2010 | Forest | Yes | | | | | Yes |
| 22 | Fox et al., 2011 | | | | Yes | | | |
| 23 | Franco et al., 2021 | Crop | | | | | | Yes |
| 24 | Gao et al., 2008 | Crop | | | | Yes | | |
| 25 | Gao et al., 2010 | Crop | Yes | | | | | |
| 26 | Gathumbi et al., 2002 | Agroforestry | Yes | | | | | |
| 27 | Ghahremani et al., 2021 | Crop | | Yes | | | | |
| 28 | Gomes et al., 2014 | Crop | | | | | | Yes |
| 29 | Gong et al., 2020 | Crop | | | | | | Yes |
| 30 | Grossiord et al., 2013 | Forest | | | | Yes | | |
| 31 | Guo et al., 2019 | Forest | | | Yes | | | |
| 32 | Han et al., 2022 | Crop | Yes | | | | | Yes |
| 33 | He et al., 2022 | Crop | | | | | Yes | |
| 34 | Hussain et al., 2023 | Crop | Yes | | | | Yes | |
| 35 | Jahansooz et al., 2007 | Crop | Yes | | | | | Yes |
| 36 | Jakhar et al., 2015 | Crop | | | Yes | | | Yes |
| 37 | Jiang et al., 2007 | Forest | | | Yes | | | |
| 38 | Jonard et al., 2008 | Forest | Yes | | | | | |
| 39 | Khan & Mcvay, 2019 | Crop | Yes | | | | | |
| 40 | Kherif et al., | Crop | | | | | | Yes |

| | | | | | |
|---|---|---|---|---|---|
| | 2023 | | | | |
| 41 | Khokhar et al., 2021 | Crop | | Yes | |
| 42 | Li et al., 2016 | Forest | Yes | | |
| 43 | Li et al., 2019 | Grass | Yes | | |
| 44 | Li et al., 2020 | Crop | Yes | | Yes |
| 45 | Li et al., 2021 | Crop | | | Yes |
| 46 | Liu et al., 2013 | Crop | Yes | | |
| 47 | Liu et al., 2019 | Grass | | | Yes |
| 48 | Liu et al., 2021 | Forest | Yes | | |
| 49 | Luo et al., 1999 | Forest | Yes | | |
| 50 | Luo et al., 2004 | Forest | | | Yes |
| 51 | Ma et al., 2019 | Crop | Yes | | |
| 52 | Ma et al., 2020 | Crop | Yes | | |
| 53 | Ma et al., 2022 | Grass | | | Yes |
| 54 | Machiwal et al., 2021 | Crop | | Yes | |
| 55 | Mao et al., 2012 | Crop | Yes | | |
| 56 | Mbanyele et al., 2021 | Crop | | | Yes |
| 57 | Mohsenabadi et al., 2008 | Crop | | | Yes |
| 58 | Moore et al., 2011 | Forest | Yes | | |
| 59 | Mu et al., 2013 | Crop | | | Yes |
| 60 | Nagase & Dunnett, 2012 | Grass | | Yes | |
| 61 | Nelson et al., 2018 | Crop | Yes | | |
| 62 | Niu et al., 2018 | Grass | | | Yes |
| 63 | Nyawade et | Crop | Yes | | Yes |

| | | | | | | | | |
|---|---|---|---|---|---|---|---|---|
| | al., 2019 | | | | | | | |
| 64 | Ogindo & Walker, 2005 | Crop | Yes | | | | | |
| 65 | Ouyang et al., 2017 | Crop | | | Yes | | | |
| 66 | Ouyang et al., 2018 | Crop | | | Yes | | | |
| 67 | Pankou et al., 2021 | Crop | | | | | | Yes |
| 68 | Powell & Bork, 2004 | Agroforestry | Yes | | | | | |
| 69 | Rahman & Ye et al., 2017 | Crop | Yes | | | Yes | Yes | Yes |
| 70 | Rahman et al., 2017 | Crop | Yes | | | Yes | Yes | Yes |
| 71 | Ren et al., 2019 | Crop | Yes | | | | | Yes |
| 72 | Ren et al., 2021 | Crop | Yes | | | | | |
| 73 | Schume et al., 2004 | Forest | Yes | | | | | |
| 74 | Shang et al., 2022 | Crop | Yes | | | | | Yes |
| 75 | Shen et al., 2023 | Crop | | | | | | Yes |
| 76 | Shu et al., 2014 | Grass | Yes | | | | | |
| 77 | Singh et al., 2020 | Crop | | | Yes | | | |
| 78 | St Aime et al., 2020 | Crop | Yes | | | | | Yes |
| 79 | Su et al., 2018 | Grass | | | | | | Yes |
| 80 | Sun et al., 2014 | Crop | Yes | | | | | |
| 81 | Te et al., 2023 | Crop | | | | | | Yes |
| 82 | Tetteh et al., 2019 | Agroforestry | Yes | Yes | | | | |
| 83 | Thomas et al., 2021 | Forest | Yes | | | | | |
| 84 | Wan et al., 2021 | Crop | | | | | Yes | |
| 85 | Wan et al., | Grass | Yes | | | | | |

| No. | Reference | Type | | | | | | |
|---|---|---|---|---|---|---|---|---|
|  |  | 2022 | | | | | | |
| 86 | Wang & Chen, 2015 | Crop | | | Yes | | | |
| 87 | Wang & Wang, 2016 | Grass | Yes | | | | | |
| 88 | Wang P et al. 2022 | Agroforestry | | | | Yes | | |
| 89 | Wang et al., 2008 | Forest | | Yes | | | | |
| 90 | Wang et al., 2011 | Crop | | | Yes | | | |
| 91 | Wang L et al., 2015 | Agroforestry | | Yes | | | | |
| 92 | Wang H et al., 2015 | Grass | | | | | Yes | Yes |
| 93 | Wang X et al., 2016 | Crop | | | Yes | | | |
| 94 | Wang J et al., 2016 | Grass | Yes | | | | | |
| 95 | Wang et al., 2017 | Forest | | | | | Yes | |
| 96 | Wang et al., 2020 | Crop | Yes | | | | | |
| 97 | Wang W et al., 2022 | Forest | | | | | Yes | |
| 98 | Wang et al., 2023 | Crop | Yes | | | | | |
| 99 | Wu et al., 2015 | Crop | | | Yes | | | |
| 100 | Wu et al., 2016 | Grass | | Yes | | | | |
| 101 | Xiong et al., 2016 | Crop | | | | | Yes | |
| 102 | Xu Z et al, 2022 | Grass | | | | | | Yes |
| 103 | Xu et al., 2008 | Grass | Yes | | | | | Yes |
| 104 | Xu et al., 2019 | Agroforestry | | | | Yes | | |
| 105 | Xu et al.,2008 | Grass | Yes | | | | | Yes |
| 106 | Xu et al., 2023 | Crop | Yes | | | | | Yes |
| 107 | Xu W et al., | Grass | | | | | | Yes |

| No. | Reference | Land use type | | | | | | | |
|-----|-----------|---------------|---|---|---|---|---|---|---|
| | 2022 | | | | | | | | |
| 108 | Yang et al., 2023 | Forest | Yes | | | | | | |
| 109 | Ye et al., 2015 | Crop | Yes | | | Yes | | | Yes |
| 110 | Yun et al., 2021 | Forest | Yes | Yes | | | | | |
| 111 | Žalac et al., 2023 | Agroforestry | Yes | | | | | | Yes |
| 112 | Zhang et al., 2005 | Forest | | Yes | | | | | |
| 113 | Zhang et al., 2008 | Grass | | | | | | | Yes |
| 114 | Zhang et al., 2017 | Crop | Yes | | | | | | |
| 115 | Zhang et al., 2017 | Grass | | | | | | | Yes |
| 116 | Zhang et al., 2021 | Grass | | | | | | | Yes |
| 117 | Zhang et al., 2022 | Crop | | | | | Yes | | |
| 118 | Zhang et al., 2022 | Grass | | | | | Yes | | |
| 119 | Zhang et al., 2022 | Forest | | | | | | Yes | |
| 120 | Zhao X et al., 2012 | Crop | Yes | | | | | | |
| 121 | Zhao Y et al., 2012b | Agroforestry | | | Yes | | | | |
| 122 | Zhao et al., 2016 | Forest | Yes | | | | | | |
| 123 | Zhao et al., 2021 | Forest | Yes | | | | | | |
| 124 | Zhao et al., 2022 | Crop | | | Yes | | | | |
| 125 | Zhao Y et al., 2012a | Crop | Yes | | | | | | |
| 126 | Zheng & Chen, 2017 | Crop | | | Yes | | | | Yes |
| 127 | Zhou et al., 2019 | Crop | Yes | | | | | | |
| 128 | Zhou et al., 2023 | Crop | Yes | | | | | | |

| 129 | Zhu et al., 2023 | Crop | Yes |
|------|------------------|------|-----|
| 130 | Zuo et al. 2008 | Grass | Yes |

Note: SWC: soil water content (cm$^3$ m$^{-3}$); IR: steady infiltration rate (mm min$^{-1}$); RO: runoff (mm); E: soil evaporation (mm day$^{-1}$); LT: leaf transpiration (mmol m$^{-2}$ s$^{-1}$); Th: throughfall (mm): WUE: water use efficiency (g m$^{-2}$ mm$^{-1}$)